# Factors Regulating Fluid Restitution and Plasma Volume Reduction over the Course of Hemodialysis

**DOI:** 10.3390/toxins15010031

**Published:** 2022-12-31

**Authors:** Jen-shih Lee, Lian-pin Lee

**Affiliations:** 1Global Monitors, Inc., San Diego, CA 92067, USA; 2Department of Biomedical Engineering, University of Virginia, Charlottesville, VA 22908, USA

**Keywords:** hemodialysis, fluid restitution, filtration coefficient, permeability fraction, plasma volume reduction

## Abstract

Over the course of hemodialysis, fluid and protein are restituted from the tissue compartment to the circulation compartment through the endothelia. Our previous model analysis on fluid and protein transport during hemodialysis is expanded to account for changes occurring in the tissue. The measured initial and end plasma protein concentration (PPC, Cp and Cp’) for six hemodialysis studies are analyzed by this expanded model. The computation results indicate that the total driving pressure to restitute fluid from the tissue to the circulation ranges from 5.4 to 20.3 mmHg. The analysis identifies that the increase in plasma colloidal osmotic pressure (COP) contributes 78 ± 6% of the total driving pressure, the decrease in microvascular blood pressure 32 ± 4%, the increase in the COP of interstitial fluid −6 ± 3%, and the decrease in interstitial fluid pressure −5 ± 2%. Let this ratio (Cp’ − Cp)/Cp’ be termed the PPC increment. The six HDs can be divided into three groups which are to have these PPC increments 25.7%, 14.5 ± 2.6(SD)% and 8.3%. It is calculated that their correspondent filtration coefficients are 0.43, 1.29 ± 0.28 and 5.93 mL/min/mmHg and the relative reductions in plasma volume (RRPV) −22.1%, −13.1 ± 6% and −9.4%. The large variations in PPC increments and RRPV show the filtration coefficient is a key factor to regulate the hemodialysis process.

## 1. Introduction

Over the course of hemodialysis (HD), ultrafiltrate is being extracted out of the circulation by the dialyzer while interstitial fluid restituted from the interstitial fluid compartment (IFC) via the semipermeable endothelia to the circulation compartment. These two fluid movements lead to a reduction in plasma volume of the circulation, an increase in plasma colloidal osmotic pressure (COP) and a decrease in microvascular blood pressure. In the meantime, the fluid restitution from IFC can lead to these three changes: an increase in the COP of the interstitial fluid, a decrease in interstitial fluid pressure, and a reduction in lymphatic return. These three changes are dependent on three tissue factors: (i) The ratio of tissue volume to the plasma volume, (ii) the elasticity of the extracellular matrix and (iii) the lymphatic retention fraction. The model analysis developed previously is expanded for us to examine how the three tissue factors alter the rate of fluid restitution and the relative reduction in plasma volume (RRPV) over the time course of HD [1].

In our previous analysis, the movement of fluid and protein across the endothelia are governed by four circulation factors: (i) the filtration coefficient of the endothelia lining the surface of the microcirculation KSo, (ii) the reflection coefficient σ, (iii) the permeability fraction γ which specifies the protein concentration of restituted fluid as a fraction of plasma protein concentration (PPC), and (iv) the autonomous constant ζ which specifies how the microvascular blood pressure is adjusted by the autonomous system and the circulation itself as the patient is taking HD. If the patient has a larger KSo, the analysis reveals that more fluid would be restituted into the plasma compartment and lesser volume reduction would be imposed to the circulation. To recognize this beneficial effect, we term the filtration coefficient as a facilitator for the patient to have more fluid restituted from the tissue to the circulation. In this paper, we will make sensitivity analyses to examine what roles are played by the circulation and tissue factors in having more fluid restitution, lesser blood volume reduction and lesser chance to the development of intradialytic hypotension (IDH) for patients with chronic kidney disease (CKD) taking HD.

The equations governing the HD process form a self-contained system. This is to say when the initial conditions are given and the values of modelling parameters are assigned, one can use the equations to calculate the time course of variables such as the plasma volume, the PPC, the microvascular pressure and the COP of interstitial fluid. In the case that the PPC at the end of HD is measured, we can use the analysis to find a value of KSo. In six sets of hemodialysis studies, the initial and end PPC (Cp, Cp’), initial and end hematocrit (Ha, Ha’) are reported [2,3,4]. Their PPC increments (=(Cp’ – Cp)/Cp’) are in the range of 8.3% to 25.7%. In this paper, we will evaluate and examine what circulation and tissue factors of the patient group cause such a large change in PPC increments. The driving pressures to produce fluid restitution, the volume of fluid restituted and RRPV are to be calculated for the six HDs. Results comparisons will help us to gain a better understanding on how the circulation and tissue factors regulating or changing fluid restitution and plasma volume as the patient is undertaking HD. The understanding can be the base to setting up a HD protocol that can reduce the chance of IDH development.

## 2. Analyses

### 2.1. The Interstitial Fluid and Plasma Compartment

The interstitial fluid space of the entire body is simulated as one interstitial fluid compartment (IFC) with Vt(t) as its volume at time t. It is filled by an interstitial fluid having Ct(t) as its protein concentration at time t. The plasma space in the circulation is set as the plasma compartment which has Vp(t) as its volume. The plasma flowing within has Cp(t) as its PPC. (For the sake of simplicity, we also use these representations: Cp, a concentration without (t), as the initial PPC and Cp’ as the end PPC. When clarification is called for, we use Cp(0) or Cp0 instead of Cp). These two compartments are joined by endothelia (the broken line in Figure 1) which line up the entire surface of the microcirculation.

On the plasma compartment, the vascular pressure driving fluid to filtrate across the endothelia is in the range of 80 to 3 mmHg [5]. Thus the filtration flux across the microcirculation surface will be non-uniformly distributed. For a 13 generation vascular tree, it is calculated that the average of the varying pressures has a value similar to a pressure within the capillaries. This average is set as Pmic and used to represent the hydrostatic pressure in the Starling Hypothesis that induces filtration [1]. The plasma COP πp(t) is the second pressure on the vascular side to drive fluid across the endothelia.

In the cyclic hemorrhage reinfusion experiment (CHRE), it is shown that the arterial and venous blood pressure decrease linearly with the hemorrhaged volume. LaForte et al. [6] found that the decrease in arterial blood pressure of conscious rabbits is much larger than that of anesthetized rabbit. The changes in arterial and venous blood pressure lead to a change in the microvascular pressure Pmic. These changes are integrated to produce the following equation:Pmic(t) − Pmic0 = ζ (Vp(t) − Vp0)/Vp0(1)
where ζ is termed the autonomous constant to reflect the controlling role of the autonomous system on the arterial and venous blood pressure and the controlling role of arterioles on how the changes of these two pressures affect a change in Pmic(t). In Equation (1), the script 0 is added to show that pressure or volume is the one at the initiation of HD.

The interstitial fluid is embedded within an elastic extracellular matrix. A decrease in interstitial fluid volume calls for a volume reduction of the matrix and a reduction in the pressure of interstitial fluid. We simulate the pressure-volume relation as:Pt(t) − Pt0 = ξ (Vt(t) − Vt0)/Vt0(2)
where ξ is termed the elastic constant of the extracellular matrix.

As the concentration Ct(t) and Cp(t) are viewed as uniformly distributed throughout their corresponding compartment, the total protein mass in the interstitial and plasma compartment at time t are calculated as Ct(t)Vt(t) and Cp(t)Vp(t), respectively.

### 2.2. Transvascular Fluid and Protein Movements

The Starling’s hypothesis on fluid movement is used to define the transvascular fluid movement developed over the course of HD. Before HD, we have the initial transvascular filtration Jf(0) as:Jf(0) = KSo[Pmic(0) − Pt(0) + σ(πp(0) − πt(0))] = Qly(0)(3)
where Qly(0) is the rate of interstitial fluid being returned to the circulation via the lymphatic system at time 0. The second equality in Equation (3) corresponds to no volume change for the plasma compartment. After the initiation of HD, the fluid flux from the plasma compartment to IFC is given as:Jf(t) = KSo{Pmic(t) − Pt(t) + σ[πp(t) − πt(t)]}(4)

The addition of the lymphatic return Qly(t) to the fluid flux above forms the following restitution fluid flux:Jr(t) = −Jf(t) + Qly(t) = −Jf(t) + Jf(0) + Qly(t) − Qly(0)= KSo{σ[πp(t) − πp(0) − πt(t)+πt(0)] − [Pmic(t) − Pmic(0) − Pt(t)+Pt(0)]}+Qly(t) − Qly(0)(5)

This term Jr(t) is depicted as the big arrow in Figure 1 and the term Jf(0) or Jf0 as the small arrow. The lymphatic return Qly and the extraction rate Je are depicted on the right hand side of Figure 1. Over the HD period ΔT, we have the extraction volume ΔVe as Je∙ΔT.

We define the volume of fluid restituted from IFC to the plasma compartment Vr(t) and the volume of lymphatic retention Vly(t) as:(6)$$\mathrm{Vr}(t)=\int_{0}^{t}\mathrm{Jr}\left(t\right)\mathrm{dt}$$
(7)$$\mathrm{Vly}(t)=\int_{0}^{t}\left[\mathrm{Qly}\left(t\right)-\mathrm{Qly}\left(0\right)\right]\mathrm{dt}$$

Under this definition of Vr(t), we have its initial value Vr(0) as 0. We also set the end value Vr(ΔT) as ΔVr. Similarly the end value Vly(ΔT) is set as ΔVly.

As fluid is being restituted from IFC, the interstitial fluid pressure is reduced. This pressure reduction may lead to a reduction in lymphatic return [7]. Thus Qly(t) may be lesser than Qly(0) or Jf(0). As some fluid is being filtrated from the plasma compartment to IFC at the rate of Jf(0), a volume Vly(t) of that fluid is being retained by IFC over the period (0, t). Thus we term Vly(t) as the lymphatic retention volume up to time t.

In the above derivations, the transvascular flux Jf(t) is divided into two terms: Jr(t) and Jf(0). This division allows us to express Equation (5) in terms of pressure differences such as Pt(t) − Pt0. Since this difference is the one being calculated by the analysis, the form of Equation (5) allows us to do calculations without the need to set values for Pmic(0), Pt(0) and πt(0).

Let the protein concentration of the restituted fluid be Cr(t). The protein mass being restituted from IFC over the time period (0, ΔT) is calculated as the term on the left hand side of the following equation. Then, it is expressed as the term on the right hand side:(8)$$\int_{0}^{\Delta T}\mathrm{Cr}\left(t\right)\mathrm{Jr}\left(t\right)\mathrm{dt}=\gamma \;\mathrm{Cp}0\;\Delta \mathrm{Vr}$$
where γ Cp0 is the filtration weighted average of Cr(t) over the entire course of HD. The fraction γ is termed the permeability fraction. For transient calculations, the integral will be made over the period (0, t) and the value of Cr(t) is set as:Cr(t) = γ Cp(t)(9)

Protein is carried out of IFC by the fluid restitution at this rate Jr(t)Cr(t). The lymphatic return carries the protein out of IFC at the rate of Ct(t)Qly(t). Let the protein concentration of the filtration Jf0 be Cf(t). The corresponding protein flux entering the IFC is given as Jf0 Cf(t). The sum of the first two items and the subtraction of the third one form the net mass flux of protein leaving the IFC at time t.

In conjunction to the setting of Equation (9), the following equation [1] is used for the value determination of Cf(t):Cf(t) = γ^½^ Cp(t)(10)

On the Graphic Abstract of this paper, the arteriole, capillary and venule are used to represent the plasma compartment of Figure 1 and their surface that of the endothelia. From Figure 2 of Ref. [1] on the pressure distribution along a microvascular network, it is estimated that the filtration over the small arteries and arterioles at the beginning of HD can be 5.15 units while the fluid absorption over the venules and small veins −3.15 units. Their sum is the lymphatic flow. At the end of HD, the increase in plasma COP and the decrease in vascular blood pressures elevate the filtration to 6.05 units and the absorption to −13.2 units. Their difference with the subtraction of lymphatic flow is applied to reduce the volume of interstitial fluid. These filtrations and absorptions are depicted by the four big arrows shown in the Graphic Abstract. A lighter color is used for the absorption arrows to indicate that the protein concentration of absorption fluid is lower than that of interstitial and filtration fluid.

### 2.3. Changes in Volumes and Protein Mass of the Plasma Compartment and IFC

As ΔVr and ΔVly are the volume of fluid added to the plasma compartment and ΔVe is the volume of ultrafiltrate extracted, we can express the change in the plasma volume from Vp(0) to Vp(ΔT) as:ΔVp ≡ Vp’ − Vp = ΔVr + ΔVly − ΔVe(11)

The terms in Equations (9) and (10) are used to determine the protein mass added to the plasma compartment through the restitution and lymphatic retention. The extraction takes no protein out of the circulation. On the conservation of protein mass of the plasma compartment, we have
(Cp + ΔCp)(Vp + ΔVp) = Vp Cp + γ Cp ΔVr − γ^½^ Cp ΔVe + Ct ΔVly(12)

Equations (11) and (12) can be combined to yield these two equations:ΔCp/Cp’ = ΔVe/Vp − (1 − γCp/Cp’) ΔVr/Vp − (1 − γ^½^Cp/Cp’) ΔVly/Vp(13)
(1 − γ Cp/Cp’)ΔVp/Vp = −ΔCp/Cp’ + γCp/Cp’ ∙ (ΔVe/Vp)+(γ − γ^½^)(Cp/Cp’)ΔVly/Vp(14)

Equation (13) is termed the protein equation. The three terms on its right hand side describe how ΔVe, ΔVr and ΔVly contribute to form the PPC increment ΔCp/Cp’. If the PPC, γ, Ct0/Cp’ and ΔVly/Vp are known for a HD, the protein equation can be used to calculate the normalized restitution volume ΔVr/Vp. Then, Equation (14) is used to calculate the RRPV ΔVp/Vp.

On the IFC, its volume change can be written as:dVt(t)/dt = −Jr(t) − Qly(t) + Qly(0)(15)

The protein mass in the tissue compartment at time t is increasing at this rate: d[Vt(t) Ct(t)]/dt. The protein mass is leaving the IFC as they are carried in or out by restitution, filtration flux Jf0 and lymphatic return. The mass conservation for IFC is expressed as:d[Ct(t)Vt(t)]/dt = −γ Cp(t) dVr(t)/dt − Ct(t)Qly(t) + γ^½^ Ct(t)Qly(0)(16)

In view of the presence of Vp(0), Vt(0) and ΔVly in Equations (12)–(16), we use the following tissue volume ratio *φ* and lymphatic retention ratio ψ to define the roles played by the IFC on fluid and protein movement:φ = Vt(0)/Vp(0)   ψ = ΔVly/Vp(0)(17)

### 2.4. Factors Regulating the HD Process

The hemodialysis process is prescribed by these nine independent variables: Vp(t), Cp(t), Vt(t), Ct(t), Vr(t), Cr(t), Cf(t), Pmic(t) − Pmic0, and Pt(t) − Pt0, define the state of patient taking HD. They are governed by nine equations: Equations (1), (2), (5), (9), (10) and (13)–(16). The colloidal osmotic pressure πp(t) and πt(t) are regarded as dependent variables as they are calculated from Cp(t) and Ct(t) through the Landis-Pappenheimer Equation [8].

From the governing equations, we find that the fluid restitution and plasma volume reduction are governed by three groups of parameters. The first group is made up of these two HD factors ΔVe (or ΔVe/Vp) and ΔT. They define how the HD is done. The circulatory factors are KSo, σ, γ and ζ. These parameters *φ*, ψ and ξ are termed the tissue factors. Among the latter two groups, KSo, σ, γ and φ are recognized as passive factors while ζ, ξ and Ψ as active factors. The term “active” means their value can be changed by the autonomous, circulatory and lymphatic system.

### 2.5. The Reference HD and Calculation Procedures

The reference HD is set to have ΔT as 240 min and ΔVe as 2167 mL. As the initial conditions, we set Cp0 as 7.12 g/dL, Vp0 as 2785 mL and Ct0 as 2.16 g/dL. On the governing factors of the reference HD, the filtration coefficient KSo is set as 0.82 mL/(min mmHg), σ as 1, γ as 0.09, ζ as 36.7 mmHg, φ as ∞ and ψ as 0.  The last two settings are identified as one with zero tissue factor (TF0). For sensitivity analyses, calculations are made with one of the factors to vary over a range of values. In particular, calculations are also made with these filtration coefficients 0.43 and 5.88 mL/min/mmHg. The HD associated with these two KSo and the reference HD are classified as low, high and medium KSo group, respectively.

For HD that the initial and end PPC are reported, one can use Equations (13) and (14) to calculate ΔVr/Vp and ΔVp/Vp0. The calculations are made with γ set 0.09 and ψ as 0.

### 2.6. Characteristics of Six HDs

The HD experiments done by Minutolo et al. [2] to patients with normal and anemic hematocrit are identified as HD1 and HD2. The HD with high ultrafiltration rate done by Schneditz et al. [4] is identified as HD3 and the three HD done by Pstras et al. [3] at different extraction volume as HD4, HD5 and HD6. The body weight (BW) and the blood volume Vb are listed in the 2nd and 3rd column. The blood volume of HD3 is given by the reference while the others are calculated as 80 mL/kg ∙ BW. The whole body hematocrit Hw is set as 0.85∙Ha with Ha being the reported arterial hematocrit. The plasma volume is calculated as Vb(1 − Hw). They are listed in the 4th and 5th column of Table 1. The HD period ΔT, the extraction volume ΔVe, and ΔVe/Vb are given in the last three columns.

The relative extraction in reference to the plasma volume, measured initial and final PPC, the calculated PPC increment, the initial and final arterial hematocrit Ha and Ha’, and the hematocrit increment are listed in Table 2.

Calculations of the time dependent variables of HD1-6, such as plasma volume reduction and PPC increase, are done with these parameter settings: σ as 1, γ as 0.09, ζ as 36.7 mmHg, φ as 4 and ψ as 4 mmHg.  An iterative procedure is employed to find the value of KSo such that the predicted end PPC matches with the measured one.

### 2.7. Parameter Settings of Five HD Simulations

For HD1 and HD2, 48% and 49% of the blood volume is extracted out by the dialyzer over a HD time of 240 min. On the other hand, 18.4% of the blood volume is extracted out of patients in the HD3 group over a period of 20 min. The HD of simulation S3 is set to extract 50% of the blood volume out of patients of HD3 over a period of 240 min. The HD characteristics, initial PPC and calculated final PPC of patients in S3 group are listed in the last row of Table 1 and Table 2.

Because the arterial hematocrits of patients in HD1, HD2 and HD3 groups are different from each other (*p* < 0.05), the plasma volumes of HD1, HD2 and HD3 groups are also different from each other (Table 1). Although the value ΔVe/Vb for HD1, HD2 and S3 is set as about 50%, the relative extraction in reference to the plasma volume ΔVe/Vp take these values: 75%, 68.5% and 65.5%, respectively. The HD that is set to extract 68.5% of plasma volume out of patients in the HD1 group is identified as simulation S2A and that to extract 65.5% of the plasma volume out of patients in the HD1 group is identified as S3A. Other parameters of S2A and S3A are identical to that of HD1.

Simulation S2B will be the HD1 to which the filtration coefficient is changed to have 0.88 mL/min/mmHg as its value. Similarly S3B will be HD1 whose patients are to have 5.88 mL/min/mmHg as their filtration coefficient. These two values of KSo are those derived for patients taking HD2 and HD3.

## 3. Results

### 3.1. Fluid Movements and the PPC Increment

The PPC increment (ΔCp/Cp’) defines how the PPC is being concentrated or diluted by the HD process. The protein equation (Equation (13)) identifies that the PPC increment is determined by these three fluid movements: extraction by the dialyzer ΔVe, restitution of fluid from the IFC ΔVr and lymphatic retention ΔVly. The values of ΔVe/Vp, −ΔVr/Vp and ΔVly/Vp are the PPC increments generated by the three movements when the protein concentration of the latter two fluids is set as zero. The reference HD has γ set as 0.09. These two quantities (1 − γCp/Cp’) and (1 − γ^½^Cp/Cp’) for the reference HD now take 92.5% and 75% as their respective values. In reference to the protein equation, we see that the sum of these increments ΔVe/Vp0, −0.925ΔVr/Vp0 and 0.75ΔVly/Vp forms the actual PPC increment generated by the three fluid movements. It is noted that a negative increment means that ΔCp is negative, i.e., the protein concentration is being reduced or diluted. 

### 3.2. The HD Time ΔT and the Filtration Coefficient KSo

The effect of changing HD time (ΔT) on the PPC, the driving pressure Δπp − ΔPmic and RRPV is shown in Figure 2. Calculations are made for a HD time of 60, 120, 160, 200 and 240 min. Other parameters used in the calculations have the same values as that of the reference HD. As the total extraction volume remains the same, the change in ΔT from 60 to 240 min corresponds to a change of rate of extraction from 36 to 9 mL/min. The three curves depicted in Figure 2 are generated with the setting of the filtration coefficient KSo as 0.43 (identified as Low KSo), 0.88 (Medium KSo) and 5.88 (High KSo) mL/min/mmHg.

Comparisons of the three curves reveal that the higher is KSo, the lower are the PPC increment and the driving pressure. The results shown in Figure 2C indicate that more fluid restitution is generated for patients with higher KSo. Consequently the analysis projects a lesser RRPV for patients with higher KSo (Figure 2D). In particular, as the HD time is shortened from 240 min to 60 min, the RRPV drops from 2.5% to −4.6% for high KSo group and from −22% to −42% for the low KSo group.

The following special feature is shown for patients with high KSo. When the HD is run at a HD period of 240 min, the restitution rate after the first 30 min becomes larger than the extraction rate. As a result, the patient experiences an plasma volume increase of 2.5% Vp0 at the end of HD. For patients taking the 60 min HD, the significant increase in Jr(t) as shown in Figure 2C is still smaller than the rate increase in extraction. As a result, the patient is to experience a RRPV of −4.6% by the end of HD. This direction change of RRPV does not show up for patients with lower KSo.

### 3.3. Reflection Coefficient σ, Permeability Fraction γ and Autonomous Constant ζ

The permeability fraction γ sets the PPC of restituted fluid Cr as γCp. As γ takes a larger value, the restituted fluid is to have a larger protein concentration. Its addition to the plasma compartment dilutes the PPC at a lesser degree so that the plasma COP is to stay at a larger value. This increase in plasma COP leads to a higher restitution flux. Thus γ can be considered as a facilitator to enhance the restitution flux. The increasing trend in RRPV for patients having larger γ is indicated by the three curves depicted in Figure 3B.

How the RRPV changes for patients with a larger autonomous constant ζ is shown in Figure 3C. As one can see ζ functions as a facilitator for patients with low and medium KSo. For the reference HD with high KSo and low ζ, the plasma volume first decreases by 2% in about 20 min. Then, it rises to 1.025 times of Vp0 by the end of HD. Equation (1) indicates that change in Pmic will be proportional to the change in plasma volume. The calculations reveal that average of the change in Pmic over the HD period is about 0.01 mmHg. The average change is reduced to 0.04 mmHg for patients having 50 mmHg as ζ. These small changes could only generate a small change in RRPV. Accordingly the small change is revealed by the horizontal dotted line shown in Figure 3C.

### 3.4. Tissue Volume Ratio φ, Tissue Elastic Constant ξ and Lymphatic Retention Ratio ψ

The time courses on PPC increment, the driving pressure on the vascular side, that on the side of IFC, and RRPV are shown in Figure 4 for four settings of tissue factors ξ and *φ*. The first setting is the HD with zero tissue factors (TF0). The other three settings are identified as ξ15φ5, ξ15φ2 and ξ30φ2. Here, ξ15 means that the value of ξ is set as 15 mmHg and φ2 means that φ is set as 2, i.e., the volume of IFC is set as 2 times of the plasma volume Vp. The conditional change from TFO to ξ15φ5, ξ15φ2 and ξ30φ2 means the patient is changed to one having a more rigid extracellular matrix and/or a smaller IFC volume. The results shown in Figure 4 reveal that the conditional change leads to a decrease in vascular driving pressure Δπp − ΔPmic, an increase in tissue driving pressure Δπt − ΔPt, an increase in PPC increment and a decrease in RRPV to a more negative value. The decrease in RRPV means that being more rigid and/or having smaller volume can inhibit the development of more fluid restitution over the course of HD.

To assess how the lymphatic retention ratio affects the HD process, we calculate the case that ψ is set as −5%, φ as 4 and ξ as 4 mmHg. The initial and end PPC remain to have the same value. It is calculated that KSo takes 0.95 mL/min/mmHg as its value, ΔVr/Vp 69.2%, ΔVp/Vp −12.2% and DP’ 8.7 mmHg. The correspondent values calculated for the condition TF0 are 0.88 mL/min/mmHg, 66.1%, −11.7% and 8.9 mmHg respective. The value comparison on ΔVp/Vp indicates that the lymphatic retention acts as an inhibitor to the development of more fluid restitution.

### 3.5. Paired γ and Kso

In this “paired” analysis, we set the value of γ as 0, 0.2, 0.3 and 0.4 and then iterate the corresponding value of KSo that makes Cp’ the same as that of the reference HD. Five pairs (γ = 0, KSo = 0.7 mL/min/mmHg), (0.09, 0.82), (0.2, 1.03), (0.3, 1.36) and (0.4, 2.06) are derived. In Figure 5A, the rises of PPC over the course of HD for four pairs of (γ, KSo) are shown. For patients having a higher γ, the difference between the PPC and the protein concentration of restituted fluid is smaller. This smaller difference and the larger restitution flux shown in Figure 5A lead to a projection from Equation (13) that a slower rise in PPC is to occur for patients having a larger γ and KSo.

If the PPC at 2 h is measured, then we can select the right pair of (γ, KSo) such that the predicted PPC can match the initial, midway and end PPC. Previously, the initial and end PPC are used to determine a value for KSo as the value of γ is already set as 0.09.

Suppose that the HD is done over a shorter period (e.g., a ΔT of 180 min). Then, the PPC calculated from the analysis will rise in the way depicted in Figure 5B. The matching of the predicted end PPC with the measured one can serve as a support on the validity of the model analysis and the way to determine the value of γ and KSo.

The time courses of restitution flux Jr(t), the driving pressure Δπp(t) − ΔPmic(t), and the protein flux Cr(t) Jr(t) are plotted in Figure 6 for four pairs of (γ, KSo). As shown in the figures, a patient having a larger KSo; his/her restitution flux is smaller. A smaller restitution flux will lead to lesser dilution in PPC, larger rise in plasma COP, lesser reduction in plasma volume and lesser reduction in interstitial fluid pressure. As a result, the driving pressure is larger for patients with a larger filtration coefficient (Figure 6B). The time courses of the protein flux Cr(t)Jr(t) for the four pairs of (γ, KSo) are plotted in Figure 6C. It shows the largest protein flux for patients of this pair (0.4, 2.06) and no protein flux for this pair (0, 0.7). The difference between any two protein fluxes is due primarily to the difference between their correspondent γ’s.

### 3.6. Analytic Results of Six HDs and One Simulation

For the six HDs, the computed values of KSo, ΔVe/Vp, Δπp, ΔPmic, Δπt and ΔPt are listed in Table 3. The relative plasma volume reduction normalized by the blood volume are tabulated in the last column of Table 3. It is seen that the filtration coefficient of the HD1 has the lowest value 0.43 mL/min/mmHg and that of HD3 has the highest value 5.93 mL/min/mmHg. The filtration coefficients of the remaining four HDs appear to be significantly different from these two extremes. We group them as the HDM group with M to stand for the word “medium”.

The relative restitution volume ΔVr/Vp, the driving pressure DP’ (=Δπp − ΔPmic + Δπt − ΔPt) restituting fluid to the plasma compartment, and the four individual pressures normalized by DP’ are given in Table 4. One sees that the increase in plasma COP induced by the extraction and restitution and the decrease in microvascular pressure induced by the reduction in plasma volume contribute 111% of the total driving pressure and the increase in tissue COP and the decrease in interstitial fluid pressure contribute −11%.

From the values given in Table 2, Table 3 and Table 4, we calculate the average ± SD of the four HDs in the HDM group and tabulate them as the third row of Table 5. The correspondent values of KSo, ΔCp/Cp’, ΔVp/Vp and DP’ of HD1 and S3 are tabulated in the 2nd and 4th row of the table. As shown in the third column, the relative extraction of the three groups are comparable in value. On the other hand, the values of KSo, ΔCp/Cp’, ΔVp/Vp and DP’ of these three groups appear to be significantly different from each other.

Let ΔVp1 be the blood volume change calculated by the van Beaumont Hematocrit Equation [9]. The values of ΔVp1/Vb for the six HDs are those listed under −ΔHa/Ha’ of Table 2. The ΔVp1/Vb so calculated are plotted against the ΔVp/Vb listed in the last column of Table 3 in Figure 7. The linear fit of the six data points indicate that ΔVp1/Vb has a strong correlation with ΔVp/Vb. We also use the linear fit to calculate ΔVp1*/Vb from a given ΔVp/Vb and determine the standard deviation of ΔVp1/Vb and ΔVp1*/Vb. The coefficient of variation (the standard deviation/the average of ΔVp1/Vb) of the six HDs takes 24% as its value.

### 3.7. Comparisons of HD1 with Four Simulations

The time course of PPC increment, driving pressure and plasma volume for HD1, S2A and S3A are plotted in Figure 8. The patient taking S2A or S3A has the same characteristics as that of patient taking HD1. The exception is that their relative extraction is changed from 75% to 68.5% and 65.5%.

For HD1, S2B and S3B, the time courses of ΔCp(t)/Cp(t), Δπp(t) − ΔPmic(t) and Vp(t)/Vp0 are depicted in Figure 9. These HDs are done to patients of HD1. Patients in these three groups have their filtration coefficients as 0.43, 0.88 and 5.93 mL/min/mmHg, respectively. A comparison of the curves in Figure 8 and Figure 9 reveals that the change in filtration coefficient has a much larger effect on the hemodialysis process than the change in relative extraction. This observation suggests that the significant drop in PPC increments found for HD1, HD2 and S3 (or HD3) are due primarily to the increase in KSo and secondary to the decrease in ΔVe/Vp.

## 4. Discussion

In our previous analysis we examine primarily on how fluid is restituted from the tissue to the plasma compartment and how the volume of plasma compartment and the PPC are altered by the fluid restitution and ultrafiltrate extraction. The computed results indicate that the larger is the filtration coefficient, the more is the volume of fluid being restituted and the lesser is the volume reduction of the plasma compartment. Accordingly we regard the filtration coefficient works as a facilitator that can lead to more fluid being restituted from the tissue to compensate for the blood volume reduction induced by hemodialysis. The current modelling analysis indicates that these governing parameters: the permeability fraction, the reflection coefficient and the autonomous constant can also be termed as a facilitator to the development of more fluid restitution. On the other hand, a smaller volume for IFC, a more rigid extracellular matrix that contains interstitial fluid and more lymphatic retention by the IFC will cause lesser fluid restitution from the IFC to the plasma compartment. Thus it is appropriate to term these three tissue factors as an inhibitor to the generation of more fluid restitution.

The role of filtration coefficient in getting the PPC to be more concentrated and in preventing the development of hypovolemia is prominently displayed by the analytic results derived for HD1, S2A and S3A and that derived for HD1, S2 and S3. For the first group, the relative extraction drops from 75% to 68.5% and then to 65.5% while the patients have the same filtration coefficient. The correspondent PPC increment drops from 26% to 24% and 22% while RRPV from 22% to 21% and then to 18%. For the second group, the filtration coefficient drops from 0.43 to 0.88 and then 5.93 mL/min/mmHg while the relative extraction stays at 75%. The correspondent PPC increments drops from 26% to 17% and then to 7% while the RRPV from 22% to 12% and then 1%.

For the first group, the relative extractions ΔVe/Vp are selected such that their relative extraction ΔVe/Vb is about 50%. The projections of different Cp(t) and Vp(t) for this group means that ΔVe/Vp is a more direct index to characterize the effect of extraction on the HD process. This is consistent with the recognition that seven of the eight governing equations do not explicitly contain these two parameters Ha and Vb. If HD1-3 were done at the same ΔVe/Vp and ΔT [2,4], then the difference in PPC increment ΔCp/Cp’ can be linked more directly to the changes in filtration coefficient, permeability fraction, etc.

By setting the hematocrit in the microcirculation as the arterial hematocrit and the splenic RBC release as zero, van Beaumont developed a hematocrit equation to show that this RRPV ΔVp1/Vb is equal to the hematocrit increment −ΔHa/Ha’. The protein equation derived from current analysis is used to calculate ΔVp/Vb from the measured PPC increments of HD1-6. The calculations are done under the setting: γ as 0.09 and the lymphatic retention ψ as zero. As shown in Figure 7, these two RRPVs are strongly correlated (R2 = 0.96). More measurements need to be made and analyzed such that the calculation errors introduced by setting γ as 0.09, ψ as zero, the microvascular hematocrit as the arterial hematocrit and no splenic RBC release can be gauged and used to generate a methodology to assess the change in plasma and blood volume over the course of HD.

As reported by Fogh-Andersen et al. [10], the protein concentration of lymph Cly is 30% of PPC. For steady state, this result suggests that the protein concentration in IFC Ct is 30% of PPC and 30% of the protein in plasma is allowed to permeate across the endothelia [1]. In applying this percentage to the fluid being restituted back from the IFC to the circulation, we have the protein concentration of restituted fluid Cr as 0.3 Ct or 0.09 Cp. Guyton [11] mentioned in his Textbook of Physiology that the lymphatic protein concentration is 40% of PPC. With similar reasoning, the correspondent γ can take 0.16 as its value. As examined in Section 3.5, more PPC measurements within the course of HD could be used by the analysis to identify the value of γ and KSo.

By using the isogravimetric technique and changing the plasma COP, Gamble et al. [12] identify that the reflection coefficient of rabbit submandibular gland lay between 0.79 and 1.0. Guyton [11] stated in his textbook that the volume of extracellular fluid in a young adult male of 70 kg is 20% of body weight—about fourteen liters. Eleven liters is interstitial fluid and the remaining three liter is plasma. These volume data indicate that the interstitial fluid volume is about 3.7 time of the plasma volume. Hillman et al. [13] reported that the interstitial fluid volume of two species of anuran is 171 and 154 mL/kg and the plasma volume 61 mL/kg and 40.5 mL/kg, respectively. These data correspond to an IFC volume being 2.8 (=171 mL/kg/(61 mL/kg)) and 3.9 (154 mL/kg/(40.5 mL/kg)) times of the plasma volume. The compliance of anuran’s interstitial fluid space has this value 67 mL/mmHg/kg BW which corresponds to the case that ξ is to take 2.6 mmHg (=171 mL/kg/(67 mL/mmHg/kg)) as its value. In skeletal muscle, the interstitial compliance was 13 and 20 mL/kg/mmHg in hypothyroid and euthyroid rats and the corresponding numbers for the back skin are 27 and 50 mL/kg/mmHg [14]. If we set the plasma volume of these rats as 70 mL/kg [15], then we find the value of ξ for the skeletal muscle and back skin is in the range of 1.4 mmHg [=(70 mL/kg)/(50 mL/mmHg/kg)] to 5.4 mmHg [=(70 mL/kg)/(13 mL/kg/mmHg)]. The data analysis of six HD is done with *φ* set as 4 and ξ set as 4 mmHg.

Minutolo et al. [2] reported that the patients taking HD1 have their systolic blood pressure to decrease from 138 ± 23 to 118 ± 23 mmHg and the diastolic blood pressure 86 ± 9 mmHg to 75% ± 7 mmHg. Similar drops are found for patients taking HD2. The analysis of the protein data of HD1 and HD2 indicates that the microvascular blood pressure can drop by 3 to 5 mmHg by the end of HD. The determination of the drop in Pmic is done with the autonomous constant ζ set as 36.7 mmHg. If the CHRE is performed on patients and the arterial and venous blood pressure are measured, we will be able to get a more precise estimate on ζ, the reduction in microvascular pressure and the reduction in plasma volume. The reduction in plasma volume can serve as an index to specify how autonomous system is to change the state of patients over the course of HD.

## 5. Concluding Remarks

The analysis identifies seven modelling parameters or factors that control or regulate the fluid restitution and plasma volume reduction occurring over the course of HD. By analyzing the PPC data obtained from patients having normal, anemic and more anemic hematocrit, we identify that their filtration coefficient takes these values: 0.43 to 0.88 and 5.93 mL/min/mmHg while the volume of fluid restituted from tissue is 71%, 83% and 60% of the extraction volume. More PPC measurements with HD done on various protocols are called for. Their data analysis will provide more information on how the modelling parameters regulate or govern the HD process.

The data analysis of HD1 reveals that the plasma volume to the end of HD is reduced by 22%. This large reduction suggests that the patients in the HD1 group are likely to develop severe hypovolemia and subsequently IDH. Patients with CKD have large drop in arterial blood pressure during HD and a large blood pressure variability (BPV) is associated with high mortality [16]. The analysis of HD experiments to be performed to CKD patients prone to develop IDH or having large BPV may allow us to gain a better understanding on the roles played by the modelling parameters on why these patients are prone to develop IDH or to have high BPV. The analysis of the measured PPC would generate a better understanding on the role the modelling parameters or factors played in the development of IDH and/or the occurrence of a large variation in blood pressure. More knowledge on the actions and impacts of these parameters to the HD process will be useful to the design of a HD protocol that can minimize the chance for the CKD patient to develop IDH and/or to have a large BPV during hemodialysis.

## Figures and Tables

**Figure 1 toxins-15-00031-f001:**
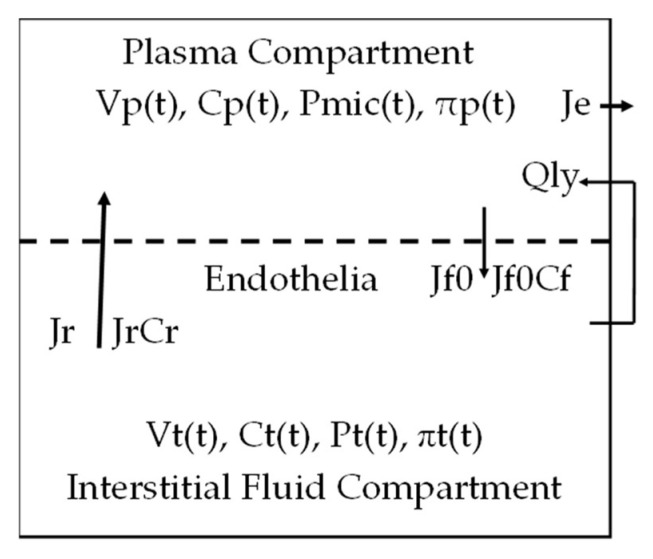
The fluid and protein movements between the interstitial fluid compartment and plasma compartment.

**Figure 2 toxins-15-00031-f002:**
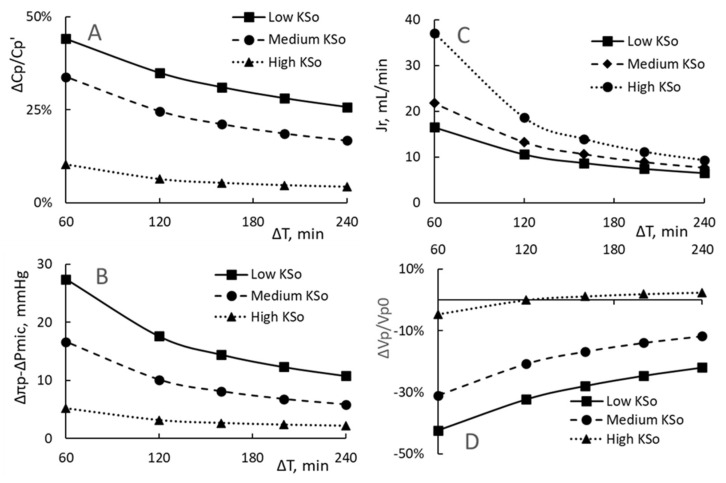
The changes in end PPC increment ΔCp/Cp’ (**A**), driving pressure Δπp − ΔPmic (**B**), the restitution flux Jr (**C**) and relative reduction in plasma volume ΔVp/Vp0 (**D**) as the HD time ΔT is shortened. The curves are calculated for patients having low, medium or high filtration coefficient.

**Figure 3 toxins-15-00031-f003:**
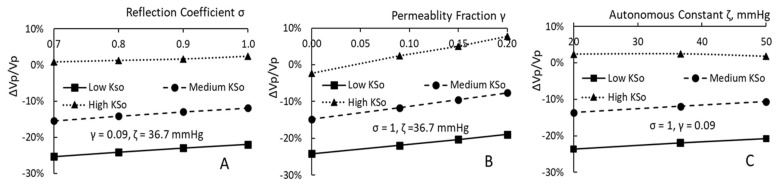
The rising trends of RRPV for patients having larger σ (**A**), γ (**B**) or ζ (**C**).

**Figure 4 toxins-15-00031-f004:**
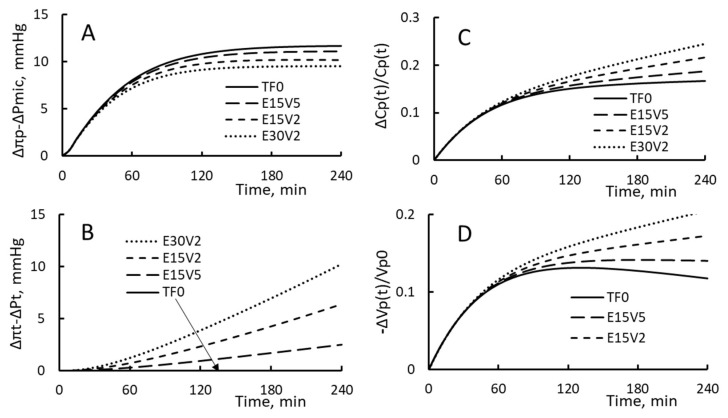
(**A**) The increase in the driving pressure on the vascular side, (**B**) The increase in the driving pressure on the interstitial side, (**C**) The increase in PPC increment and (**D**) The change in RRPV.

**Figure 5 toxins-15-00031-f005:**
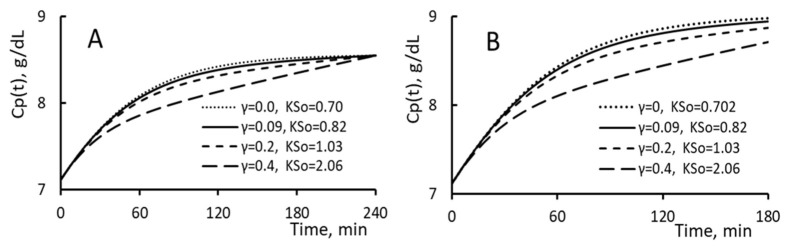
(**A**) The time course of PPC increment and (**B**) the decrease in relative plasma volume Vp(t)/Vp0 for patients having the (γ, KSo) listed in the figure.

**Figure 6 toxins-15-00031-f006:**
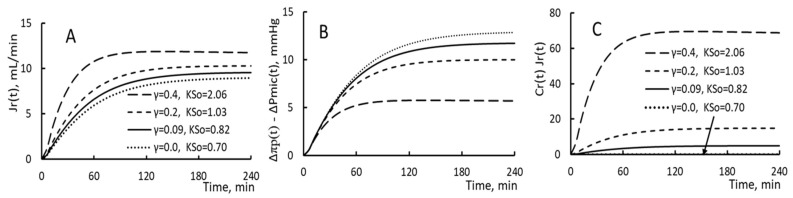
(**A**) The time course of restitution flux Jr(t), (**B**) That of driving pressure and (**C**) That of protein flux Cr(t) Jr(t).

**Figure 7 toxins-15-00031-f007:**
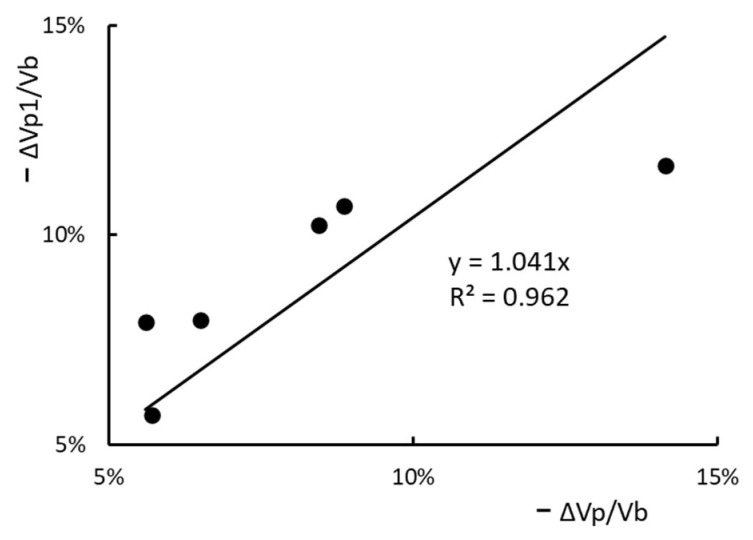
The correlation of two RRPVs for HD1 to HD6. ΔVp/Vb is one calculated from the measured PPC increment and ΔVp1/Vb is calculated from the measured hematocrit increment.

**Figure 8 toxins-15-00031-f008:**
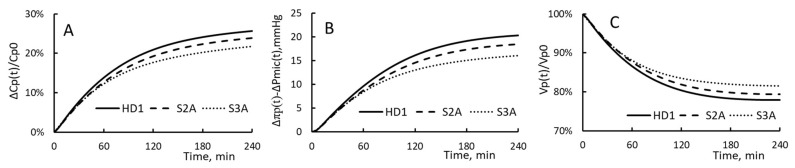
(**A**) The time course of PPC increment, (**B**) The time course of the driving pressure and (**C**) The time course of plasma volume for HD1 and S2A and S3A. Their relative extraction ΔVe/Vp is 75%, 68.5% and 65%, respectively and their filtration coefficient is 0.43 mL/min/mmHg.

**Figure 9 toxins-15-00031-f009:**
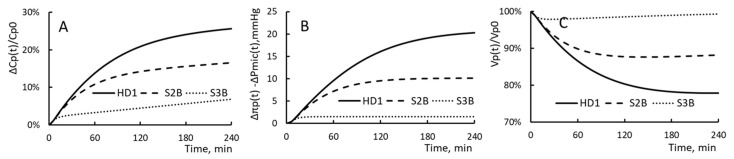
(**A**) The time course of PPC increment, (**B**) The time course of the driving pressure and (**C**) The time course of plasma volume for HD1 and S2B and S3B. Their relative extraction ΔVe/Vp is 75% and their filtration coefficient is 0.43, 0.88 and 5.93 mL/min/mmHg, respectively.

**Table 1 toxins-15-00031-t001:** Patient Characteristics of Six Hemodialysis.

	BW, kg	Vb, mL	Hw	Vp, mL	ΔT, min	ΔVe, mL	ΔVe/Vb
HD1	54.4 ± 4.1	4352	36.04%	2785	240	2089	48.0%
HD2	56.2 ± 6.4	4496	28.48%	3216	240	2203	49.0%
HD3	69.1 ± 15.9	5200	23.72%	3967	20	730	14.0%
HD4	73.1 ± 16.9	5851	29.91%	4101	237	3000	51.3%
HD5	71.2 ± 19.0	5698	30.62%	3953	237	2333	40.9%
HD6	70.5 ± 17.4	5638	29.45%	3977	236	2070	36.7%
S3A	69.1	5200	23.72%	3967	240	2600	50.0%

The data are mean ± SD. Data of HD1 & HD2 are from Minutolo et al. [3], HD2 from Schneditz et al. [9] and HD4 to 6 from Pstras et al. [5].

**Table 2 toxins-15-00031-t002:** The Relative Extractions, Changes in PPC and Hematocrit of HD1–6 & S3.

	ΔVe/Vp	Cp, g/dL	Cp’, g/dL	ΔCp/Cp’	Ha, %	Ha’	−ΔHa/Ha’
HD1	75.0%	7.12	9.58	25.7%	42.35 ± 3.5	47.94%	−11.7%
HD2	68.5%	7.29	8.69	16.1%	33.53 ± 2.4	37.35%	−10.2%
HD3	18.4%	6.89	7.51	8.3%	27.9 ± 3.1	30.3%	−7.9%
HD4	73.2%	6.43	7.76	17.1%	35.19 ± 4.0	39.41%	−10.7%
HD5	59.0%	6.50	7.49	13.2%	36.02 ± 4.3	39.14%	−8.0%
HD6	52.0%	6.58	7.44	11.6%	34.65 ± 3.2	36.75%	−5.7%
S3	65.5%	6.89	7.60	9.3%	27.9		

The arterial hematocrit of HD1 is significantly higher (*p* < 0.05) than the hematocrit of others except HD5 and that of HD3 significantly lower (*p* < 0.05) than the hematocrit of other groups.

**Table 3 toxins-15-00031-t003:** Filtration Coefficient, RRPV and Changes in Driving Pressures of Six HD.

	KSo	ΔVp/Vp	Δπp	ΔPmic	Δπt	ΔPt	ΔVp/Vb
HD1	0.43	−22.1%	16.4	−5.2	0.7	−0.5	−14.1%
HD2	0.88	−11.8%	8.9	−3.1	0.8	−0.6	−8.5%
HD3	5.93	−7.3%	3.6	−2.1	0.1	−0.1	−5.6%
HD4	1.35	−12.6%	7.6	−3.2	0.7	−0.6	−8.9%
HD5	1.44	−9.4%	5.6	2.4	0.6	−0.5	−6.5%
HD6	1.48	−8.1%	4.9	−2.1	0.5	−0.4	−5.7%
S3	5.93	−1.3%	2.8	−0.4	08	−0.6	−1.0%

The unit of KSo, Jr’, and pressures are mL/(min mmHg), mL/in and mmHg, respectively.

**Table 4 toxins-15-00031-t004:** Relative Volume of Fluid Restitution, Total Driving Pressure and Four Pressures.

	ΔVr/Vp	DP’, mmHg	Δπp/DP’	−ΔPmic/DP’	−Δπt/DP’	ΔPt/DP’
HD1	52.9%	20.3	80%	26%	−3%	−3%
HD2	56.7%	10.7	83%	29%	−7%	−5%
HD3	11.1%	5.4	66%	38%	−2%	−2%
HD4	60.5%	9.6	80%	34%	−7%	−6%
HD5	49.6%	7.0	81%	34%	−8%	−7%
HD6	44.0%	6.1	81%	35%	−8%	−7%
AVG ± SD			78 ± 6%	32 ± 4%	−6 ± 3%	−5 ± 2%
S3	64.3%	1.7	164%	20%	−47%	−37%

**Table 5 toxins-15-00031-t005:** Parameter Values of HD1, HDM and S3.

	KSo, mL/min/mmHg	ΔVe/Vp	ΔCp/Cp’	ΔVp/Vp	DP’, mmHg
HD1	0.43	75.0%	25.7%	−22.1%	20.3
HDM	1.29 ± 0.28	63 ± 9%	14.5 ± 2.6%	−13.1 ± 6%	8.3 ± 2.2
S3	5.93	65.5%	6.7%	−1.3%	1.7

## Data Availability

Data shown in the figures can be obtained with an email to mlp7@cox.net.

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
