# Peer review of "Factors Regulating Fluid Restitution and Plasma Volume Reduction over the Course of Hemodialysis"

_toxins, 2022, doi:10.3390/toxins15010031_

Round 1

Reviewer 1 Report

Minor changes suggested: 

haΔ the same values as that of the reference HD.     Line 262

should this be "had"?

in Fig. are generated with the setting of the filtration coefficient KSo as 0.43 (iden-        Line 265

Please specify the figure

How are the data points in Figure 8 and 9 different? To the observer, these graphs are identical, although the y-axis is different in Fig 8a vs Fig 9a. 

Line 446-7: please cite reference 16 or other as appropriate.

patients prune to develop IDH or having large BPV may allow us to gain a better under-      Line 506

Suggest you meant "prone"? 

Author Response

Line 262   It is changed to "have". 

Line 265. It is identified as Fig. 2. 

The correct figure for Fig. 8 is put in. 

Line 446-447.  These two references (11, 14) are added to the manuscript. 

Line 506 "prune" is changed to "prone". 

Reviewer 2 Report

- Originality/Novelty: The question is original and well defined investigating the forces which allow fluid transfer during HD session. The results are significant from the physiological point of view and may be some implication in therapeutics. There are also some older studies of the some team in this area which offers them some advantages.

- Significance: The results are well interpreted appropriately but I have one concern regarding study no.3. The data from there as they are mentioned are not really consistent with all other studies included in analysis. The duration and volumes are not consistent. It's true, that the original paper is not very clear about this but I think that we can extract some data from the original paper to integrate in this one to have consistency in data in tabel-1.  The results are well presented and organized, offer an interesting and new perspective about forces which conduct transfer in HD

- Scientific Soundness: the study is a simulation based on 6 studies, correctly designed and technically sound OK. The conclusions are consistent with data available and in the expected range. 

There are sufficient details to allow another researcher to reproduce the results. 

Interest to the Readers: The conclusions interesting for the readership of the Journal not only for nephrologist but also other physicians interested in fluid transfer in the human body. 

- Overall Merit: The work provides an advance towards the current knowledge in the dialysis principles and explanation of the fluid transfer. It can be used in the future to improve prescription and evaluation of the HD sessions.

- English Level: the English language is appropriate and understandable.

Author Response

Many thanks for the assessments given by Reviewer 2.  I hope the publication of this manuscript can stimulate more researchers to measure the time course of plasma protein concentration under various hemodialysis protocol and then use our analysis to assess fully the factors regulating the fluid restitution and reduction in plasma volume.  Then we will have more data to demonstrate whether hypovolemia is the culprit inducing the patient to develop intradialytic hypotension.